# *Candida albicans* Morphology-Dependent Host FGF-2 Response as a Potential Therapeutic Target

**DOI:** 10.3390/jof5010022

**Published:** 2019-03-05

**Authors:** Sandeep Vellanki, Eun Young Huh, Stephen P. Saville, Soo Chan Lee

**Affiliations:** South Texas Center for Emerging Infectious Diseases (STCEID), Department of Biology, The University of Texas at San Antonio, San Antonio, TX 78249, USA; sandeep.vellanki@my.utsa.edu (S.V.); eunyoung.huh@utsa.edu (E.Y.H.); stephen.saville@utsa.edu (S.P.S.)

**Keywords:** angiogenesis, FGF-2, morphogenesis, Candidalysin

## Abstract

Angiogenesis mediated by proteins such as Fibroblast Growth Factor-2 (FGF-2) is a vital component of normal physiological processes and has also been implicated in contributing to the disease state associated with various microbial infections. Previous studies by our group and others have shown that *Candida albicans*, a common agent of candidiasis, induces FGF-2 secretion in vitro and angiogenesis in brains and kidneys during systemic infections. However, the underlying mechanism(s) via which the fungus increases FGF-2 production and the role(s) that FGF-2/angiogenesis plays in *C. albicans* disease remain unknown. Here we show, for the first time, that *C. albicans* hyphae (and not yeast cells) increase the FGF-2 response in human endothelial cells. Moreover, Candidalysin, a toxin secreted exclusively by *C. albicans* in the hyphal state, is required to induce this response. Our in vivo studies show that in the systemic *C. albicans* infection model, mice treated with FGF-2 exhibit significantly higher mortality rates when compared to untreated mice not given the angiogenic growth factor. Even treatment with fluconazole could not fully rescue infected animals that were administered FGF-2. Our data suggest that the increase of FGF-2 production/angiogenesis induced by Candidalysin contributes to the pathogenicity of *C. albicans*.

## 1. Introduction

*Candida albicans* is a commensal/opportunistic fungal pathogen most commonly associated with mucosal diseases in humans. Lethal infections by *C. albicans* are continuously increasing in parallel with the growing proportion of vulnerable individuals such as immunocompromised patients and/or patients with indwelling medical devices [1,2,3]. In particular, disseminated infections pose a serious threat as mortality rates can exceed 40% even when patients receive antifungal therapy [4].

Many of the antifungal drugs that are currently used in clinical practice target fungal cells by inhibiting their growth or by killing them; they include azoles (targeting ergosterol synthesis), polyenes (physiochemically targeting ergosterol), and echinocandins (targeting cell wall synthesis) [5]. However, like many other microbial pathogens, *C. albicans* can develop resistance against antifungal drugs via (1) mutations in drug target genes, (2) up-regulation of multidrug resistance genes, or (3) the Hsp90-mediated stress response pathway [6,7]. Indeed, many clinical *C. albicans* isolates are multidrug resistant, and the emergence of drug resistance often leads to poor outcome in the treatment [8]. Therefore, it is imperative to develop novel therapeutic options that are not prone to resistance by the fungus.

Angiogenesis is the development of new blood vessels from pre-existing vessels [9]. It is regulated by proteins that either activate or inhibit the process [10]. Fibroblast Growth Factor-2 (FGF-2) is a pro-angiogenic protein that promotes angiogenesis in an autocrine fashion [11,12]. Studies have shown that FGF-2 is a more potent inducer of angiogenesis then other pro-angiogenic proteins such as Vascular Endothelial Growth Factor (VEGF) [13]. While modulation of FGF-2 or angiogenesis has become a common target in the treatment of cancer [14,15,16], studies from Ben-Ami et al. have shown such therapies could potentially be extended to treat fungal infections [17]. When *Aspergillus fumigatus* infected mice were treated with FGF-2 alone or in combination with antifungals, there was a significant increase in survival rate compared to the untreated group. However, the role of angiogenesis in *C. albicans* infection and disease progression is understudied.

Ashman et al. have shown that *C. albicans* increases angiogenesis in a murine model of systemic candidiasis [18]. Our group and others have revealed in vitro that *C. albicans*–mammalian host cell interactions result in increased production of proangiogenic growth factors, including FGF-2, and expression of the gene encoding VEGF [19,20,21]. However, the underlying molecular mechanisms of how the fungus induces FGF-2/angiogenesis in hosts remain elusive. 

This study identifies a fungal factor that modulates host FGF-2 secretion and explores the role of FGF-2 in *C. albicans* infections. We found that in vitro induction of endothelial cell FGF-2 production is dependent on *C. albicans* morphology, with an increase in FGF-2 protein secretion only noted when the fungus was present in the filamentous form. Moreover, we determined that Candidalysin, a secreted hypha-specific toxin, regulates this process. Finally, using a murine model of systemic candidiasis, we discovered that treatment of *C. albicans* infected animals with FGF-2 results in increased mortality, suggesting that *C. albicans* induces FGF-2/angiogenesis to enhance pathogenicity.

## 2. Materials and Methods

### 2.1. Ethics Statement

All animal experiments were performed in strict accordance with the guidelines of the University of Texas at San Antonio (UTSA) Institutional Animal Care and Use Committee (IACUC) and in full compliance with the United States Animal Welfare Act (Public Law 98-198) and National Institute of Health guidelines. The animal protocol used in this study was approved by the UTSA IACUC under protocol MU104. The experiments were conducted in the Division of Laboratory Animal Resources (DLAR) facilities that are accredited by the Association for Assessment and Accreditation of Laboratory Animal Care (AAALAC).

### 2.2. Cell Culture

Primary Human Umbilical Vein Endothelial Cells (HUVECs) were purchased from Lonza and were seeded into a T75 flask and maintained at 37 °C + 5% CO_2_ in Endothelial Cell Basal Medium containing hydrocortisone, abscorbic acid, Insulin Growth Factor, heparin, and FBS (EBM; Lonza) according to manufacturer instructions. The components of the HUVEC medium that were not added during the experiments include gentamicin and two proangiogenic growth factors, FGF-2 and VEGF, that are often used to culture HUVEC. Confluent flasks of cells were trypsinized and seeded into a 96-well plate for challenge with *C. albicans*.

### 2.3. Fungal Strains and Growth Conditions

*C. albicans* strains were maintained as glycerol stocks at −80 °C and were propagated by streaking on yeast–peptone–dextrose (YPD) agar plates as needed. A day before the infection (as explained below), colonies of cells from the plate were transferred to YPD liquid medium and were incubated overnight (16 h) at 30 °C with shaking (180 rpm). The yeast cells were then washed with sterile phosphate-buffered saline (PBS), resuspended, and diluted to the desired concentration using sterile PBS. The list of strains used in this study is shown in Table 1.

### 2.4. HUVECs Challenge with *C. albicans*

HUVECs were counted and the cell number was adjusted to seed 5 × 10^3^ cells/well in 100 µL medium in a 96-well plate. HUVECs were challenged with 5 × 10^4^ cells/well (MOI = 10; 10 µL/well in PBS) of each of the *C. albicans* strains for 24 h. The supernatants were collected from each well and FGF-2 Enzyme Linked Immuno Sorbent Assay (ELISA; R&D systems) was performed according to the manufacturer’s instructions to determine FGF-2 protein levels. When tetracycline-regulatable strains were used, doxycline were added to appropriate wells at a final concentration of 20 µg/mL [22]. For experiments involving compound 9029936 [23], the compound was added to appropriate wells in various concentrations as explained in Section 3.1. All ELISA measurements were performed in at least two independent experiments with three technical repeats.

### 2.5. HUVECs Challenge with *C. albicans* Spent Medium

The concentrations of *C. albicans* strain SC5314 (with or without compound 9029936) and the non-hypha-forming mutant *efg1*Δ/Δ [29] were adjusted to 5 × 10^6^/mL, and they were grown for 24 h in HUVEC cell culture medium (EBM) at 37 °C. The medium was centrifuged at 2500× *g* for 15 min and the supernatant filtered through a sterile 0.22 µm filter. The resulting medium was immediately added to HUVECs and incubated for 24 h before proceeding to the ELISA. 

### 2.6. HUVECs Challenge with *C. albicans* Nonviable (Heat-Killed and PFA-Treated) Strains

Wild-type SC5314 and *efg*1Δ/Δ mutant strains were resuspended at a final concentration of 5 × 10^6^/mL in EBM and incubated at 37 °C until germination was observed (~3 h) in the tubes containing the SC5314 strain. As a control, we also incubated both *C. albicans* strains at 28 °C (no germ tube control). The vials containing the cells were then subjected to heat (95 °C) for 30 min [32]. Since heat treatment could alter the cell wall of *C. albicans*, we also included a paraformaldehyde (PFA)-treated group. For the PFA treatment, the fungal cells were incubated in PFA for 30 min then washed with PBS several times. For the challenge experiment, either 5 × 10^4^ viable or 5 × 10^4^ nonviable (heat-killed or PFA-treated) cells were added to each HUVEC well and incubated for 24 h. Both viable and nonviable *C. albicans* were also plated on YPD plates to confirm the loss of viability in the heat- and PFA-treated groups. 

### 2.7. HUVECs Challenge with Candidalysin Peptide

Candidalysin peptide (SIIGIIMGILGNIPQVIQIIMSIVKAFKGNK) [30] was a kind gift from Dr. Julian Naglik (King’s College London). The peptide was prepared as a 10 mg/mL stock in sterile water. Serum-starved HUVECs were seeded in a 96-well plate and incubated with Candidalysin peptide diluted to 9 µM in sterile water for 24 h. Supernatants were collected to perform ELISA. 

### 2.8. Evaluation of FGF-2 Monotherapy and Combination Therapy with Fluconazole in a Murine Model of Systemic Candidiasis

Balb/C mice (male, 4–6 weeks old) were purchased from Charles River Laboratories. For *C. albicans* infections, the mice were infected with 1 × 10^6^ SC5314 yeast cells in 100 µL sterile PBS via a lateral tail vein. For monotherapy, the infected mice were given 1.6 µg of recombinant human FGF-2 (R&D systems) in 100 µL sterile PBS intravenously at 3 and 5 h after infection, as described previously [17]. For combination therapy, fluconazole was administered daily at 0.5 mg/kg via the intraperitoneal route starting 5 h after infection for 7 days. We also included animal groups infected with SC5314 alone or SC5314 treated with fluconazole. Negative controls included non-infected groups given either PBS or rFGF-2 only. Post-infection survival of the mice was monitored twice daily, and body weights were measured once daily (as this is a good indicator of mouse health). Animals which suffered a weight loss of more than 10% a day or total of 20% from the initial point were humanely sacrificed. Differences between the survival curves were evaluated for significance using the Kaplan–Meier test. The animals were housed in an SPF room, and after infection they were housed in a BSL2 room. The animals were given free access to food and sterile water. All of the mice in the same cage were part of the same treatment group.

### 2.9. Statistics

Prism (Version 7. GraphPad Software Inc.) was used to perform statistical analysis. A *p* value of ≤0.05 was considered to be statistically significant. All in vitro experiments were performed in tripicate, and each experiment was performed on at least three separate occasions. Data are expressed as mean ± SEM. In vivo challenge experiments were performed on two independent occasions with *n* = 5 in each group. Two independent experiments showed similar results and the data shown are from one representative experiment.

## 3. Results

### 3.1. FGF-2 Protein Secretion from Hosts is Dependent on the Morphology of the *C. albicans*

In the current study, HUVECs were challenged with either a wild-type (SC5314), a non-hypha-forming mutant (*efg1*Δ/Δ) (Appendix A), or a regulatable tet-NRG1 strain [26]. Nrg1 is a transcriptional repressor which suppresses the expression of hypha-specific genes in yeast-form cells. The tet-NRG1 strain was constructed by placing one allele of NRG1 under the control of a tetracycline-regulatable promoter so that the morphology could be manipulated by adding or omitting doxycycline (dox) [26]. While the addition of dox permits filamentation, omission results in a yeast-locked morphology. As shown in Figure 1a, compared to the uninfected PBS control (mock), FGF-2 production was significantly higher only when *C. albicans* was able to filament—in the wild type (WT) and *tet-NRG1* (+dox). We did not observe a substantial change in HUVECs challenged with the efg1Δ/Δ and tet-NRG1 (−dox) strains (when *C. albicans* cells remained in the yeast form). As a control, an ELISA was also performed on supernatants from wells with *C. albicans* only (no HUVECs); however, as expected, no FGF-2 protein above background could be detected (Appendix A). We also determined that the presence of doxycycline does not influence the FGF-2 response (Appendix A).

As an alternative approach to test how *C. albicans* morphology affects FGF-2 production, we challenged HUVEC cultures with the *C. albicans* WT strain SC5314 in the absence or presence of compound 9029936 for 24 h. Compound 9029936, identifed by Romo et al., blocks *C. albicans* filamentation and biofilm formation [23]. In agreement with the data presented in Figure 1a, HUVECs responded to untreated *C. albicans* (hyphae) with increased FGF-2 expression; however, the presence of the drug (2.5 to 40 µM) significantly diminished this effect (Figure 1b). Compound 9029936 by itself does not affect FGF-2 response from HUVECs (Appendix A). Taken together (Figure 1a,b), these results suggest that the host FGF-2 response is specific to the hyphal form of *C. albicans*.

### 3.2. Viable *C. albicans* Hyphae are Required for the Induction of the FGF-2 Response

We next determined whether the FGF-2 production/induction is dependent on the viability of the fungal cells. To that end, *C. albicans* WT (SC5314) were allowed to germinate by incubating at 37 °C in EBM, and then the hyphae were heat-inactivated or PFA-treated (nonviable) and added to HUVEC-containing wells for 24 h. As shown in Figure 2, while live (no heat or PFA treatment) *C. albicans* hyphae were able to induce significant FGF-2 protein production, HUVECs challenged with nonviable *C. albicans* hyphae do not display this phenomenon when compared to untreated controls (mock). We also did not observe any difference in FGF-2 secretion between HUVECs challenged with nonviable WT and those challenged with viable/nonviable *efg1*Δ/Δ. The control group which was incubated at 28 °C did not any show any germ tube formation. When the live counterparts of this group were added to HUVECs and incubated at 37 °C for 24 h, as expected, they switched to hyphal growth and induced a significant increase in FGF-2 secretion when compared to their heat-killed and PFA-treated counterparts (which cannot make hyphae due to a lack of viability). These results suggest that a factor produced from live *C. albicans* hyphae is required to elicit an FGF-2 response.

Once it was confirmed that *C. albicans* hyphae were required to elicit an FGF-2 host response, we wanted to test if hyphal-cell-wall-associated proteins are involved in this process. Als3—a member of the agglutinin-like sequence (Als) family of proteins—expressed on the hyphal surface is required for *C. albicans* adhesion and invasion of epithelial and endothelial cells [33]. Bcr1 is a transcriptional regulator of several hyphal cell wall proteins, including the Als3 invasin [34]. We tested *als3*Δ/Δ and *bcr1*Δ/Δ strains for their ability to induce FGF-2 production, but we observed no significant change when compared to their respective parental strains (Appendix A). We also tested the possible role of secreted aspartyl proteases (Saps; suggested to play a role in *C. albicans* virulence) in modulating the host FGF-2 response. To that end, HUVECs were challenged with mutant strains lacking either the Saps 1–3 or 4–6 subfamilies or with the WT strain; however, deletion of the Sap subfamilies did not impact FGF-2 production from HUVECs in a 24-hour period (Appendix A). 

### 3.3. Candidalysin Induces Host FGF-2 Protein Secretion

Candidalysin, encoded by the *ECE1* gene, is a 32-amino-acid peptide, with its dominant form being a 31-amino-acid peptide after cleavage by Kex1p [35], and was the first cytolytic toxin identified in *C. albicans* [30]. Since Candidalysin is a hypha-specific toxin and was found to be a key factor involved in activating specific host responses, we decided to test if Candidalysin also regulates the host FGF-2 response. We used several *ECE1* gene-modified strains to evaluate the relationship between Candidalysin and the host FGF-2 response. HUVECs were challenged with a *C. albicans* ECE1-null mutant in which both copies of the gene were deleted (*ece1*Δ/Δ), or with a strain in which one copy of a full-length allele of ECE1 was restored (*ece1*Δ/Δ + *ECE1*), or the strain lacking the region of ECE1 that encodes Candidalysin (*ece1*Δ/Δ + *ECE1*_Δ184-279_), or the wild-type parental strain (BWP17 + CIp30) for 24 h. As shown in Figure 3a, while the ece1Δ/Δ mutant and the ece1Δ/Δ + *ECE1*_Δ184-279_ strain failed to cause a substantial increase in FGF-2 secretion, the parental and reconstituted *ece1*Δ/Δ + *ECE1* strains induced a significant increase when compared to the uninfected control (mock). 

To verify that Candidalysin is the mediator of the FGF-2 response, we challenged HUVECs with the Candidalysin peptide and, as shown in Figure 3b, addition of the peptide alone was sufficient to induce a significant increase in FGF-2 secretion when compared to the untreated vehicle control (sterile water). Thus, the findings from Figure 3a,b confirm that Candidalysin induces secretion of FGF-2 from endothelial cells. On the other hand, it is unlikely that the host FGF-2 induces the expression of Candidalysin as wild-type SC5314 did not exhibit hyphal growth when cultured in the presence of recombinant FGF-2 (rFGF-2) (data not shown). 

To determine if the spent medium from *C. albicans* is sufficient to induce FGF-2 secretion, the *C. albicans* efg1Δ/Δ mutant and the SC5314 wild-type strain (with or without 10 µM compound 9029936) were grown in EBM for 24 h. The medium was filtered, then added to HUVECs and incubated for 24 h. However, we did not observe a significant difference in FGF-2 production among the groups (Appendix A).

### 3.4. FGF-2 Enhances Mortality Rate in a Murine Model of Systemic Candidiasis

To evaluate the role of FGF-2/angiogenesis in systemic candidiasis, we infected mice with the wild-type strain SC5314 via a lateral tail vein either alone or in addition to two FGF-2 (1.6 μg) treatments (at 3 h and 5 h post-infection). The other treatment groups were infection + fluconazole (0.5 mg/kg) treatment for seven days or infection + a combination of fluconazole and FGF-2. The infected group treated with FGF-2 showed a significant increase in the mortality rate, which is evident from all mice reaching death by Day 6, compared to the no-treatment group, in which the animals start succumbing from Day 6 onwards (Figure 4). Interestingly, co-treating the infected mice with both FGF-2 and fluconazole resulted in increased mortality rates when compared to the infected mice group treated with fluconazole alone. Once the fluconazole treatment was stopped, we observed a decreased survival rate from 100% to 80% on Day 12 in WT + fluconazole group (not shown in figure). Treatment of uninfected mice with PBS or FGF-2 did not have any impact on the mortality (survival rates) or morbidity (body weights) of the animals, indicating that administration of FGF-2 is only detrimental in the presence of an active *C. albicans* infection. 

## 4. Discussion

Understanding host–pathogen interactions provides a fundamental platform for the development of new therapeutic interventions. Our previous studies revealed an intriguing outcome of the host–pathogen interaction: the induction of the pro-angiogenic growth factor FGF-2 in mammalian hosts by fungal pathogens, including *C. albicans* [19,20]. Our findings are congruent with the study by Ashman et al. where they observed an increase in angiogenesis, characterized by enhanced endothelial cell proliferation including formations of capillary buds and small blood vessels in brain and kidney sections, in a murine model of systemic candidiasis [18]. However, the underlying mechanisms of the *C. albicans*–host FGF-2/angiogenesis response has not yet been fully explored. 

*C. albicans* can grow in several different morphologies and it is the switch from the yeast to hyphal form which is most closely linked to its capacity to cause disease. During pathogenesis, while the yeast morphology is essential for colonization and dissemination through the bloodstream, the hyphal form is required for invasion and damage to host tissues [36,37]. During host–*C. albicans* interactions, the host response has been shown to vary depending on the morphological form of the fungus [38,39,40]. This prompted us to test whether the induction of host FGF-2 secretion is also linked with a particular morphological form of *C. albicans*. To this end, we used HUVECs because they recapitulate the features of endothelial cells lining the lumen of blood vessels [41]. Moreover, HUVECs have already been extensively used to characterize the endothelial cell FGF-2 response [42] and to investigate *C. albicans*–endothelial cell interactions [43,44,45,46]. As the results in Figure 1a,b show, *C. albicans* can induce FGF-2 secretion only in the hyphal form.

Previous studies have shown that the host response may vary depending on the viability of the *C. albicans* [47,48,49]. We therefore compared the host FGF-2 response from endothelial cells challenged with both viable and nonviable hyphae. The increase in FGF-2 response was specific to live/viable hyphae (Figure 2). Fungal invasion of host cells is a hallmark of *C. albicans* infection, and Als3 is a key mediator of *C. albicans* invasion of epithelium and endothelium [33]. Our findings suggest that the invasins (and definitely Als3) are not responsible for inducing the host FGF-2 response (Appendix A). Secreted aspartyl proteases are virulence factors of *C. albicans* which are essential for degradation of host proteins and promote *C. albicans* invasion/penetration of epithelial cells [50]. However, we found that the Saps subfamilies 1–3 and 4–6 are also not involved in inducing the host FGF-2 response (Appendix A). Based on these findings (Figure 2, Appendix A), we hypothesized that a hypha-specific factor produced by *C. albicans*—after or during invasion—is involved in eliciting the FGF-2 response. 

Candidalysin is a *C. albicans* hypha-specific toxin encoded by *ECE1* [30]. Although the *ece1* mutant strains can still form hyphae and invade the host epithelium, studies have shown that these mutants fail to cause epithelial cell damage and activate pro-inflammatory signaling (danger) pathways in oropharyngeal and vulvovaginal candidiasis [30,51,52]. Studies on *A. fumigatus* show that it produces a toxin (gliotoxin) that regulates host angiogenesis [53]. This raises the possibility that toxins from other fungal systems can also potentially regulate the host FGF-2 response. Indeed, as the results in Figure 3 show, *C. albicans* induces FGF-2 secretion only when it can produce functional Candidalysin toxin. However, when spent medium from *C. albicans* hyphal cultures was added to HUVECs, we did not observe a significant change in FGF-2 secretion (Appendix A). It is possible that hyphae create an “invasion pocket” in the endothelial cell which allows for elevated microenvironmental concentrations of Candidalysin. Our results are consistent with the recent observation that epithelial cells were damaged when challenged with *C. albicans* hyphae; however, they were not damaged when challenged with the spent culture medium (discussed in [35]). Nevertheless, our results strongly support the assertion that Candidalysin is primarily responsible for eliciting the host FGF-2 response. While prior studies have been dedicated to elucidating the relationship between Candidalysin and host epithelial responses, not much was known about the regulation of endothelial cell responses by Candidalysin. Here we provide the earliest evidence that Candidalysin also plays a role in regulating/mediating endothelial cell responses. More studies will be required to determine whether Candidalysin also plays a role in any other endothelial cell responses.

*What is the consequence of the induction of proangiogenic growth factors during C. albicans infections?* Previous studies have shown that in microbial infections, host angiogenesis can either enhance host defense mechanisms or contribute to pathogenicity. For example, treatment with FGF-2 improves antifungal drug activity in a murine model of aspergillosis [17]. Also, ribonuclease 5 angiogenin is known to possess microbicidal activity against pathogenic bacteria in the gastrointestinal tract [54]. In contrast, recent findings have demonstrated that tuberculosis bacteria induce granuloma-associated angiogenesis, which contributes to its pathogenesis [55,56]. Another study found the formation of capillary buds and blood vessels around *C. albicans* foci within infected brains and kidneys [18]. Our discovery that the administration of recombinant FGF-2 into mice post-infection with *C. albicans* significantly increased mortality compared to non-FGF-2-treated animals suggests that angiogenesis mediated by FGF-2 enhances the pathogenicity of *C. albicans* during a disseminated infection.

From the current study, it appears that angiogenesis is harmful to the host in disseminated *C. albicans* infections; however, the precise mechanism by which angiogenesis enhances the pathogenicity of the fungus remains unclear. Neither blocking of FGF-2 with a neutralization antibody nor addition of recombinant FGF-2 impacts the ability of *C. albicans* to damage host endothelial cells in vitro (Vellanki and Lee, Unpublished data). We also found that addition or neutralization of cytokines such as interleukin-8 (IL-8) by a drug or antibody did not affect the endothelial cell FGF-2 response to *C. albicans* infections (Vellanki and Lee, Unpublished data) Based on these results, we hypothesize that FGF-2/angiogenesis is a direct response induced by *C. albicans* to enhance its dissemination into the deeper host tissues, as previously observed in *Mycobacterium tuberculosis* infections [57]. Our future studies will focus on elucidating the extent to which *C. albicans* hyphae and Candidalysin contribute to increasing host angiogenesis in the murine systemic candidiasis model by using existing *ece1* mutants that make hyphae but do not secrete Candidalysin and, if necessary, by constructing yeast-locked strains that produce the toxin.

The emergence of antifungal drug resistance has limited our ability to treat *C. albicans* infections and often leads to poor clinical outcomes. Thus, the identification of novel therapeutic avenues that can be exploited to treat *C. albicans* infections is required. Host-directed therapeutic approaches which enhance the host’s ability to fight the infection, rather than targeting the *C. albicans* components, have already been investigated [58]. As FGF-2 secretion/angiogenesis is apparently enhanced during a *C. albicans* infection, this process could be targeted by inhibiting FGF-2 function. Blocking a host response related to fungal pathogenicity will represent a new paradigm for treating fungal infections as they can be used irrespective of *Candida* resistance to current antifungals. Moreover, by targeting a host component of the disease process, the risk of the fungus developing a resistant mechanism to overcome this is massively reduced. Systemic *Candida* infections pose a serious risk with high mortality rates; thus, a new approach to block host FGF-2 functions during systemic infections could be an effective treatment option. There are drugs which inhibit angiogenesis and/or block FGF-2 function [14], and some of them are already approved by the FDA to treat other diseases. A successful demonstration of efficacy of any of these drugs would eliminate the time normally needed for drug development and enable them to be far more rapidly applied to patients with systemic/disseminated candidiasis. Intriguingly, during our preliminary studies, we discovered that *C. auris*—in which Candidalysin has not been identified—also enhanced the secretion of FGF-2 from host cells (Vellanki and Lee, unpublished data). This approach, therefore, could potentially be extended to disseminated infections caused by other *Candida* species.

## Figures and Tables

**Figure 1 jof-05-00022-f001:**
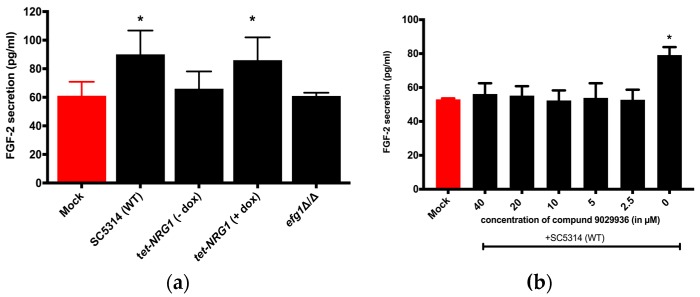
Host Fibroblast Growth Factor-2 (FGF-2) response is dependent on the morphology of *C. albicans.* (**a**) Human Umbilical Vein Endothelial Cells (HUVECs) were challenged with SC5314 (wild-type, WT), *tet-NRG1* (+/− dox), or *efg1*△/△ strains for 24 h. FGF-2 levels were measured using ELISA. One-way ANOVA was significant (*p* = 0.0005). Dunnett’s multiple comparison was used to compare each infected group with the uninfected group (mock). A *p* value of <0.05 was considered to be significant, as indicated by (*). (**b**) HUVECs were challenged with the wild-type strain SC5314 in the presence (40 to 2.5 µM) or absence (0 µM) of compound 9029936 for 24 h, and the amount of FGF-2 measured in the supernatants. One-way ANOVA was significant (*p* = 0.042). Dunnett’s multiple comparison test demonstrated a statistically significant difference (* *p* < 0.05) only between the Mock group and the infected group without compound 9029936 treatment (0 µM).

**Figure 2 jof-05-00022-f002:**
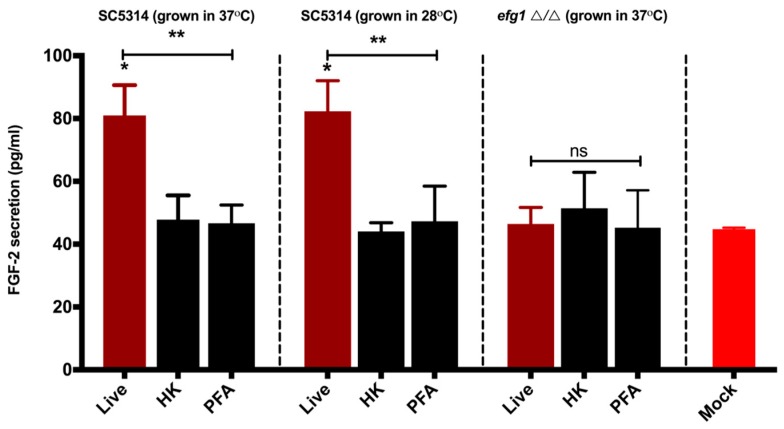
A factor(s) from viable *C. albicans* hyphae regulates the host FGF-2 response. *C. albicans* SC5314 (WT) strain was grown in Endothelial Cell Basal Medium (EBM) at 37 °C until germ tubes were visible (~3 h). Controls were SC5314 grown at 28 °C and the *efg1*△/△ mutant at 37 °C for the same time period (during which no germ tubes were detected). Each group was split into three subgroups: no treatment (live) or treatment with heat (HK) or paraformaldehyde (PFA). HUVECs were challenged with live, HK-, or PFA-treated *C. albicans* and were maintained at 37 °C for 24 h. FGF-2 levels in the culture supernatants were measured by ELISA. Both one-way ANOVA (*p* = 0.009) and Dunnett’s multiple comparison test (*p* < 0.05) showed statistically significant differences between the Mock and SC5314 live groups (as indicated by *). There was also a statistically significant difference between the live and nonviable groups (HK and PFA) in SC5314 groups (as indicated by ** for *p* < 0.05).

**Figure 3 jof-05-00022-f003:**
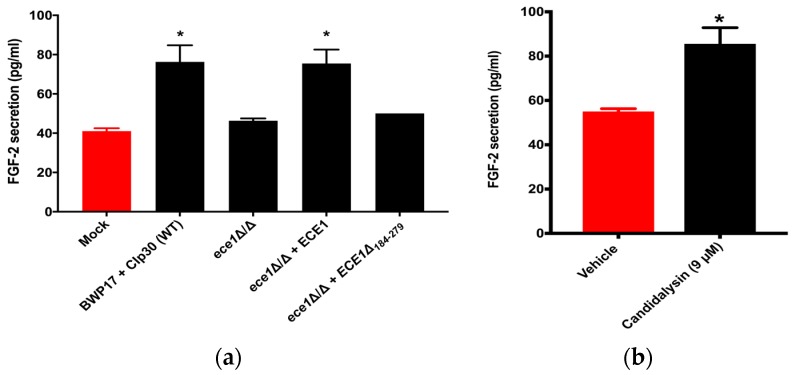
Candidalysin regulates the host endothelial FGF-2 response. (**a**) HUVECs were challenged with BWP17 + CIp30 (WT), *ece1*△/△, *ece1*△/△ + *ECE1*, or *ece1*△/△ + *ECE1*_△184-279_ for 24 h. FGF-2 levels were measured using ELISA. One-way ANOVA was significant (*p* = 0.001). Dunnett’s multiple comparison demonstrated that compared to the uninfected group (Mock), only HUVECs challenged with BWP17 + CIp30 or *ece1*△/△ + *ECE1* showed a statistically significant increase in the FGF-2 response (*p* < 0.05), as indicated by (*). (**b**) HUVECs were treated with vehicle control (water) or Candidalysin peptide for 24 h. A two-tailed *t*-test showed a statistically significant difference between the two groups (*p* = 0.049), indicated as (*).

**Figure 4 jof-05-00022-f004:**
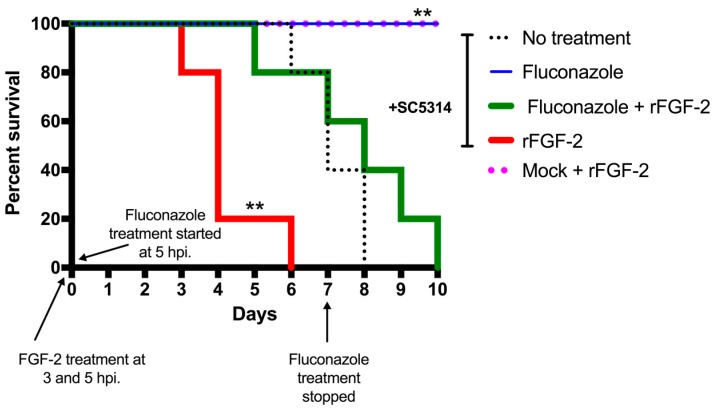
Treatment with FGF-2 increases the mortality rate in a murine model of systemic candidiasis. Balb/C mice were challenged with 1 × 10^6^ SC5314 yeast cells via a tail vein. Fluconazole (0.5 mg/kg) was administered daily via the intraperitoneal (IP) route from the day of infection for 7 days in the respective animal groups. Recombinant FGF-2 (1.6 µg each dose/mouse) was administered intravenously in the respective groups (infected and uninfected) 3 h and 5 h post-infection (hpi). A log-rank (Mantel–Cox) test was statistically significant (*p* < 0.0001). A pair-wise comparison was also performed to compare the following infected groups: no treatment vs. rFGF-2 (** *p* = 0.006), no treatment vs. rFGF-2 + Fluconazole (*p* = 0.291), and no treatment vs. Fluconazole (** *p* = 0.002), as indicated on the graph. Once the fluconazole treatment was stopped, the survival rate of the Fluconazole group changed to 80% (with the death of 1 of the mice) on Day 12 post-infection (not shown on graph). Mice injected with only PBS and rFGF-2 maintained a 100% survival rate.

**Table 1 jof-05-00022-t001:** List of strains.

Strain	Reference
SC5314	[24]
CAN33 (*efg1*Δ/Δ)	[25]
SSY50-B (*tet-NRG1*)	[26]
SN152	[27]
17322 (*als3*Δ/Δ)	[28]
*bcr1*Δ/Δ	[29]
BWP17 + CIp30	[30]
*ece1*Δ/Δ	[30]
*ece1*Δ/Δ + *ECE1*	[30]
*ece1*Δ/Δ + *ECE1*_184-279_	[30]
SAP456MS4A/B (*Sap4-6*Δ/Δ)	[31]
SAP123MS4C/D (*Sap1-3*Δ/Δ)	[31]

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
