# Peer review of "Candida albicans Morphology-Dependent Host FGF-2 Response as a Potential Therapeutic Target"

_jof, 2019, doi:10.3390/jof5010022_

Round 1

Reviewer 1 Report

Summary:

This manuscript by Vellanki, et al. seeks to identify the Candida albicans factor(s) responsible for elevating Fibroblast Growth Factor 2 (FGF-2) expression in an endothelial cell challenge model. The authors use a variety of genetically modified C. albicans strains to determine that morphogenesis is required for induction of the FGF-2 response. They then use a series of mutants to rule out roles for hyphal adhesins and secreted proteases in driving FGF-2. The authors then identify a crucial role for ECE1, and more specifically its protein product Candidalysin (a fungal toxin), in mediating FGF-2 release by using genetic and functional approaches. Lastly, the authors demonstrate the impact of their work by showing that rFGF-2 exacerbates C. albicans infection using a murine model of systemic candidiasis. Thus, it is concluded that FGF-2 elicited by C. albicans serves to amplify pathogenicity or inhibit host fungicidal activity. This article is very well written and the data largely sound. I have a few comments below that may help improve the manuscript.

Major comments:

1. Is it possible that FGF-2 amplifies expression of ECE1 (or other virulence factors, such as SAPs) in C. albicans or enhances hyphal growth? The authors may want to test this using qRT-PCR and microscopy methods.

2. Do the authors have any histological images of kidneys (or other infected organs) from infected mice treated with and without rFGF-2? It may help to conceptualize the effect of FGF-2 on pathogenesis and/or the increased invasion phenotype that is mentioned in line 330.

3. The authors mention they have treated mice with FGF-2 in the absence of infection and this led to no mortality. I think this is an important group to show in Fig. 4. It drives home the point that FGF-2 is only detrimental in the presence of active infection.

4. Lines 299-303 are awkwardly written and I’m unsure of what the authors are trying to state. It is also equally possible that hyphae are required for creating an “invasion pocket” in the endothelial cell, which allows for elevated microenvironmental concentrations of Candidalysin. I think this should be stated here. If you still feel it’s a stability issue, the authors should grow the Δ/Δece1 strain in cell culture medium, harvest it, and supplement with Candidalysin peptide for various lengths of time to assess stability/activity.

Minor comments:

1. The authors may want to capitalize the word “candidalysin” throughout, given that it is indeed a described protein.

2. Line 48: Do you mean treated with FGF-2 inhibitors? Otherwise, this is contradictory to your findings and would serve as a good discussion point. Perhaps biological site specificity or pathogen specificity?

3. There are several typos is Table 1. The plasmid used for BWP17 is CIp30 (not 31). The Candidalysin deletion mutant nucleic acid deletions should be 184-279. If you want to type the strains correctly, I believe CIp10 was used to create ece1Δ/Δ, Δ/Δece1+ECE1, and Δ/Δ+ECE1184-279 as it would restore URA3 prototrophy. Also, the Δ/Δ nomenclature should be used consistently throughout each strain description.

4. In line 88, it would be useful to put the multiplicity of infection (MOI) here.

5. Change “Als3” to “als3” in Fig. S2.

6. Line 206: Candidalysin is technically a 32 amino acid peptide. However, the dominant form is 31 amino acids due to subsequent cleavage by Kex1p after initial cleavage by Kex2p.

7. Line 213: Please correct the Candidalysin deletion mutant amino acids to 184-279.

Author Response

We attached our response letter.

Reviewer 2 Report

In their manuscript, Vellanki et al. use in vitro evidence to suggest a new role for candidalysin in promoting the expression of angiogenic factors that contribute to Candida albicans pathogenesis. It will be of general interest to the C. albicans community to note a new potential virulence mechanism linked to candidalysin. The authors also demonstrate a detrimental role for the angiogenic factor FGF-2 to host survival during murine systemic candidiasis. The figures and written arguments are generally well presented, but my enthusiasm is dimmed by technical questions relating to rationale, methods, and missing controls.

Major comments:

1.       The first results section and figure are used to argue that filamentous C. albicans cells are required for FGF-2 induction in HUVECs. However, no images are provided to establish the morphology of WT and other strains during co-culture with HUVECs. Hyphal inductions rarely produce 100% true hyphal cells and the efg1Δ is not afilamentous under strong induction conditions. The authors should provide light microscopy evidence to strengthen the argument that cultures were filamentous/afilamentous, thus filamentation is required for FGF-2 production.

a.       On a related argument, the hgc1Δ mutant is a much better candidate to support the argument that hyphal forms are required for FGF-2 elicitation. Efg1 is a transcription factor that perturbs expression of several gene expression programs not directly tied to hyphae, whereas Hgc1 is important for hyphal extension with little to no transcriptional influence.

2.       The authors need two additional controls for Figure 1:

a.       The authors should confirm that doxycycline treatment alone does not induce FGF-2 secretion as a control for their tet-NRG1 experiments.

b.      If compound 9029936 is resuspended in DMSO, the authors should address if DMSO alone plus C. albicans cells blocks secretion of FGF-2. DMSO may influence yeast stress responses and eukaryotic ROS signalling. [Sadowska-Bartosz et al. Dimethyl sulfoxide induces oxidative stress in the yeast Saccharomyces cerevisiae, FEMS Yeast Research, Volume 13, Issue 8, 1 December 2013, Pages 820–830]

c.       Further, can the authors comment on how they controlled for differences in cell count and total biomass in the hyphal and yeast cultures used to challenge HUVECs in Figure 2? Did the authors consider one filament and one yeast cell as equal during cell counting?

3.       Methods relating to the animal studies are incomplete and, as currently described, would not be reproducible in another laboratory. The Journal of Fungi has endorsed the ARRIVE guidelines for animal reporting. Authors should refer to the ARRIVE checklist (https://www.nc3rs.org.uk/sites/default/files/documents/Guidelines/NC3Rs%20ARRIVE%20Guidelines%20Checklist%20%28fillable%29.pdf) and at the minimum provide the following details:

a.       Age and sex of mice used in studies.

b.      Animal housing conditions and access to food and water.

c.       Use of lateral or central tail vein for intravenous injections?

d.      Whether the data in Figure 4 are a compilation of the two studies (all ten mice) or representative of one experiment?

e.      The rationale for rFGF-2 and fluconazole dosing and the use of recombinant human FGF-2 as opposed to recombinant murine FGF-2.

4.       Several control datasets in the manuscript are referred to as ‘not shown’. In the interests of transparency, authors should provide supplemental figures for the following data not shown:

a.       Line 151 – ELISA for FGF-2 expression in C. albicans only wells

b.      Line 245 – decrease in survival for WT + fluconazole group

c.       Line 246-247 – Negative control mice for animal studies

Minor comments

1.       Line 69 – incomplete sentence.

2.       Line 70 – change were to was

3.       Lines 300-302 – Check the sentence structure.

Author Response

We attached a response letter.

Reviewer 3 Report

This manuscript presents and discusses that hyphae form of C. albicans can induce the FGF-2 production from endothelial cells and candidalysin is a key factor to activate the production of FGF-2. The authors made a well-explained manuscript. However, some points need correction before accepting.

Although the authors showed that candidalysin peptide can induce FGF-2 production from endothelial cells, the spent medium from C. albicans could not induce FGF-2 secretion. As mentioned in the M&M part, the spent medium was from 24 hours culture of C. albicans grown in HUVEC cell culture medium without interacting with endothelial cells. However, Moyes et al., showed that C. albicans secretes candidalysin only when the fungus interacted with epithelial cells. Hence, the authors may need to use the spent medium from a culture containing both  C. albicans and endothelial cells to clarify whether candidalysin is a key factor to induce FGF-2 production. 

The authors used ECE1 gene modified strains to perform some experiments, but one strain is confusing with its name.  In the text and table 1, the authors used ece1 △ +ECE184-257 but this strain was  mentioned in Figure 3a as ece1 △ +ECE184-279. According to the reference, ece1 △ +ECE184-279 is the correct one.

line 78: Neither FGF2, VEGF or gentamycin.. should be Nether .... nor.

line 78: Where did the authors use VEGF and gentamycin?

line 79: Why do the authors trypsinize the confluent flasks? Should this be confluent flasks of cells?

line 204:there is not p-value for **.

line 234: The authors used P=0.049 instead of P < 0.05, could the authors explain the difference?

line 242: extra period was found next to "day 6". 

Two question marks were found in lines 252 and 253, and the sentences were not finished. 

Figure 4

The symbol for "No treatment" looks like bold dash-line instead of bold line.

Author Response

We attached a response letter. 

Round 2

Reviewer 2 Report

The authors have satisfactorily addressed most of my comments. The only text correction I request at this time is on lines 147-148, which refer to the efg1 mutant as yeast-locked. This mutant is capable of forming pseudohyphae, which can be seen in Figure S1, and should not be referred to as yeast-locked.

Author Response

We revised it to non-hypha-forming mutant in lines 102 and 149.

Reviewer 3 Report

The authors made a great revision and addressed all the comments, however, some small points that need correction before accepting.

line 192: "hyphal wall associated protein" should be hyphal cell wall associated protein.

line 211: the p-value of two asterisks should be <0.01. 

line 222: "while the null ece1 Δ/Δ   mutant" could be either the ECE1 null mutant or ece1 Δ/Δ   mutant.

Author Response

Line 194, we revised to 'hypha cell wall associated proteins' as suggested. 

Line 212-214, we obtained the range of the p value based on a 95% confidence setting in the PRISM software. We used this parameter to measure the statistic difference between two groups throughout the manuscript. Thus, we would like to keep using the p value <0.05 as it is. 

Line 222, we corrected as suggested.